# Supraclavicular Lymph Node Dissection in Breast Cancer with Synchronous Supraclavicular Metastases: A Systematic Review and Network Meta-Analysis

**DOI:** 10.3390/cancers17132081

**Published:** 2025-06-21

**Authors:** George Shiyao He, Jolene Li Ling Chia, Emmeline Elaine Cua-De Los Santos, Wong Hung Chew, Wee Yao Koh, Qin Xiang Ng, Samuel Ow, Serene Si Ning Goh

**Affiliations:** 1NUS Yong Loo Lin School of Medicine, National University of Singapore, Singapore 119077, Singapore; 2Department of Surgery, National University Hospital, Singapore 119074, Singapore; 3Department of Radiation Oncology, National University Hospital, Singapore 119074, Singapore; 4Saw Swee Hock School of Public Health, National University of Singapore and National University Health System, Singapore 119228, Singapore; 5Department of Haematology-Oncology, National University Cancer Institute, Singapore 119074, Singapore

**Keywords:** breast cancer, supraclavicular lymph node metastases, supraclavicular lymph node dissection, network meta-analysis, disease-free survival, overall survival, locoregional therapy

## Abstract

Patients with breast cancer who develop spread to the lymph nodes above the collarbone—known as synchronous supraclavicular lymph node metastases—face a poor prognosis. While standard treatment typically includes radiation and chemotherapy, the benefit of surgically removing these lymph nodes remains uncertain. In this study, we reviewed existing evidence and analyzed outcomes across different treatment strategies. We found that adding surgery does not consistently improve survival outcomes. However, patients with limited disease involving only the level V lymph nodes may experience a survival benefit from selective dissection. In contrast, more extensive surgical procedures were associated with worse outcomes, likely reflecting more aggressive underlying disease. These findings suggest that surgery may be selectively beneficial in patients with limited nodal involvement and should be considered on a case-by-case basis.

## 1. Introduction

Worldwide, breast cancer is one of the most prevalent cancers, accounting for approximately 12.5% of all annual cancer diagnoses [1]. Among its varied manifestations, the occurrence of synchronous ipsilateral supraclavicular lymph node metastases (sISLMs) in breast cancer patients without distant metastasis is relatively uncommon, comprising only 1% to 4% of cases [2]. The presence of sISLM serves as a significant prognostic factor, often heralding a poor prognosis. A notable proportion of patients with sISLM experiences disease progression, marked by the onset of distant metastases within a year of detection. Moreover, survival rates are disappointingly low, with a reported rate of 53.46% after 3 years [3], declining further to 5% to 34% after 5 years [4].

Historically, sISLM in breast cancer was defined as a form of distant metastasis (M1) rather than N3 under the 1988 American Joint Committee on Cancer (AJCC) TNM Staging System 3rd Edition due to poor prognosis, leading to treatment strategies primarily focused on palliation rather than curative interventions such as local surgical treatment. However, advancements in diagnosis and therapy have led to a more optimistic outlook. In 2003, the AJCC reclassified isolated tumor cells in regional lymph nodes as clinical N3c (cN3c), distinguishing sISLM as a localized condition rather than a metastatic one, thus opening the door to curative locoregional treatments [5].

The current management of sISLM typically involves multimodal therapy, featuring preoperative systemic therapy (ST) such as neoadjuvant chemotherapy (NAC) and/or hormonal therapy, followed by complete or partial breast resection, in accordance with the latest NCCN guidelines [6]. However, achieving a complete response (CR) rate of supraclavicular lymph nodes (SLNs) following NAC remains suboptimal, with reported rates hovering around 50% [7], raising significant concerns regarding residual tumor burden. Optimal locoregional strategies have been explored, with the addition of either postoperative radiotherapy (RT) or supraclavicular lymph node dissection (SLND) surgery.

Conventional RT regimens to the supraclavicular and subclavian regions are typically 50 Gy in 25 fractions of 2 Gy [6], with additional boost doses of 10 to 16 Gy in 5 to 8 fractions of 2 Gy, resulting in total cumulative doses of 60 to 66 Gy [8]. However, the current RT regimen is limited by normal tissue tolerance and challenges in achieving curative agents’ volume [9]. Escalating RT doses do not improve patient prognosis; instead, they heighten the risk of RT-related complications, including brachial plexus (BP) injury, lymphedema of the upper limb, musculoskeletal and soft tissue damage, radiation pneumonitis, and secondary neoplasms [10].

Aggressive locoregional surgery, such as SLND, may offer a more precise and comprehensive therapeutic approach when compared to RT. This strategy aims to reduce local tumor burden and prevent the spread of tumor cells through lymphatic vessels. Additionally, surgically removing the metastatic supraclavicular lymph node allows for a more accurate assessment of sentinel lymph nodes (SLNs) following NAC [9].

Comparative studies assessing patient outcomes between SLND surgery and neck RT are crucial in establishing definitive treatment guidelines. Unfortunately, the absence of comparative trials evaluating different treatment regimens poses a significant challenge in this regard. A comprehensive review of the existing literature reveals significant controversy and a lack of consensus regarding the most effective locoregional treatment strategy. Existing observational studies report significant variability and conflicting results on the efficacy of SLND versus RT, with 5-year overall survival (OS) rates ranging from 33% [11] to 78.9% [12] for patients undergoing SLND with RT and ST and from 35% [13] to 80.2% [14] for those receiving RT with ST alone. To address this gap, our study therefore aims to synthesize existing research via a systematic review and network meta-analysis (NMA) approach to evaluate oncological outcomes, particularly overall survival (OS) and disease-free survival (DFS), across three modalities: (1) SLND combined with RT and ST, (2) SLND combined with ST alone, and (3) ST alone, using RT + ST as the reference group. The findings of this analysis could offer valuable insights to guide future locoregional treatment strategies for patients with cN3c breast cancer.

## 2. Methods

This systematic review and meta-analysis adhered to the guidelines detailed in the Preferred Reporting Items for Systematic Reviews and Meta-Analyses (PRISMA) Statement [15]. The study protocol was also prospectively registered in PROSPERO (registration number: CRD42023443098).

### 2.1. Search Strategy and Selection Criteria

We searched four electronic databases—Medline (via PubMed), Embase, Scopus, and the Cochrane Central Register of Controlled Trials (CENTRAL)—from inception until 31 December 2023 for eligible English-language publications. Studies included breast cancer patients with synchronous sISLM without distant metastasis undergoing locoregional treatment. The search strategy employed combinations of key terms such as “Breast Cancer”, “Supraclavicular Lymph Node Metastasis”, “Radiotherapy”, and “Dissection.” The detailed search strategy can be found in Appendix A. Titles and abstracts of studies identified from the electronic databases were reviewed by two researchers (G.S.H. and J.L.L.C.), independently and in parallel, to determine eligibility according to the inclusion and exclusion criteria. Duplicates were identified and removed via Endnote. In cases of disagreement relating to identification or eligibility of studies, consensus was reached through discussion with input from the senior author (S.S.N.G.). The first stage involved reviewing the title and abstract while the second involved a full-text evaluation of studies selected in the first stage to determine eligibility using the same inclusion and exclusion criteria. When there was disagreement about study relevance, consensus was reached through discussions with the senior author (S.S.N.G.). The reference lists of possibly relevant studies and review articles were examined to ensure no pertinent studies were overlooked. Ongoing clinical trials were identified via ClinicalTrials.gov, and conference abstracts were included if relevant, original data were available. If necessary, study sponsors and investigators were contacted to clarify outcome definitions or obtain additional trial-level data.

Studies were included if they met the following criteria: patients with breast cancer and synchronous sISLM, as defined by each study; patients undergoing locoregional therapy, including local surgery, radiotherapy, or both; and randomized controlled trials, prospective studies, or retrospective non-randomized cohort studies. Accordingly, patients with metachronous supraclavicular lymph node metastasis; patients with distant metastasis, as defined in each study; lymphadenopathy due to other causes; case reports, case studies, reviews, commentaries, and letters to the editor; and studies published in languages other than English or without an English translation were excluded.

### 2.2. Data Extraction

Two authors (G.S.H. and J.L.L.C.) independently extracted the data using a standardized data collection form, and any conflict was resolved by discussion or input from with the senior author (S.S.N.G.). The included articles were assessed, and the data related to the search were extracted and analyzed. Study variables recorded were study characteristics (e.g., study year, study design, geographic origin, sample size), patient demographic data, molecular subtypes (HER2, hormone receptor status, and Ki67 proliferation index), diagnosis, interventions (level of neck dissections and RT regime), and outcomes. In this review, “level V dissection” refers to the posterior cervical triangle, anatomically defined as the area posterior to the sternocleidomastoid muscle, bounded by the trapezius muscle and clavicle. This follows the standard head and neck anatomical classification. However, definitions of dissection levels varied slightly across studies, and our categorization into “level V-only” versus “extended dissection” was based on the level classifications explicitly reported by the original study authors. The primary outcomes of interest were overall survival (OS) and disease-free survival (DFS) across different locoregional therapies, which were evaluated using hazard ratio (HR) and 95% confidence interval (CI) of HR.

Before performing the network meta-analyses, Guyot et al.’s graphical reconstructive approach [16] was used to recreate individual patient data (IPD) from the published Kaplan–Meier curves. The images of the Kaplan–Meier curves were digitized and processed to obtain the corresponding time values and step functions. The survival information of individual patients was recovered using numerical solutions to the inverted Kaplan–Meier product limit equations. The accuracy of the reconstructed IPD dataset was assessed through visual comparison with the original curves and by matching the reconstructed log-rank values with the initially reported values.

### 2.3. Statistical Analysis

OS and DFS outcomes were synthesized using a frequentist random-effects NMA for OS, while DFS was analyzed using standard meta-analysis techniques. The random-effects model accounted for variability between studies, including differences in patient populations and treatment protocols. The NMA was conducted using the R software (version 3.3.3) and the netmeta package. HRs and 95% CIs were extracted or calculated for each treatment comparison. Treatments were compared using a network diagram to visualize the relationships between different treatment strategies, with RT + ST as the reference group. Both direct and indirect comparisons were combined to estimate treatment effects.

### 2.4. Assessment of Heterogeneity and Inconsistency

Heterogeneity between studies was assessed using the I^2^ statistic, with values categorized as low (<30%), moderate (30–60%), and substantial (>60%). A global inconsistency test was conducted using the Q-test to evaluate inconsistency across the entire network, and local inconsistency was checked using node-splitting models to compare direct and indirect estimates.

### 2.5. Assessment of Quality of Evidence

To assess the quality of evidence for the primary outcomes, OS and DFS, we applied the GRADE (Grading of Recommendations, Assessment, Development and Evaluation) framework (gradepro.org). The GRADE system evaluates the certainty of evidence across four levels: high, moderate, low, and very low. The quality of evidence was assessed based on five domains: risk of bias, inconsistency, indirectness, imprecision, and publication bias. For each outcome, two independent reviewers (G.S.H. and J.L.L.C.) used the GRADEpro software (2025) to assign evidence ratings, with input from the senior author (S.S.N.G.) in cases of disagreement. Randomized controlled trials (RCTs) were initially considered high-quality, and non-randomized studies were considered low-quality. The quality of evidence could be downgraded if significant concerns were found in any of the GRADE domains or upgraded if the studies showed large effect sizes and dose–response relationships, or if plausible confounding factors were addressed.

### 2.6. Sensitivity Analysis

Sensitivity analyses were conducted to assess the robustness of the findings. These included analyses by dissection level (strictly level V vs. level V+) and by RT dosage (standard dose vs. higher dose). Additionally, a leave-one-out analysis was performed to assess whether any single study had a disproportionate influence on the overall results.

## 3. Results

As illustrated in Figure 1, the literature search of the four electronic databases (PubMed, Embase, Scopus, and CENTRAL) yielded 691 articles. An additional five studies were also included from reference lists of original articles’ set. After the removal of 267 duplicates, title and abstract screening was conducted, excluding a further 369 studies as they were either repeated studies, did not report on breast cancer patients with sISLM, lacked relevant locoregional therapy outcomes, their full text was unavailable, were non-English publications, and/or were case reports or literature reviews. Further analysis through full-text evaluation resulted in the exclusion of 35 additional studies. A total of 25 studies were included for systematic review, while 10 met the eligibility criteria for inclusion in network meta-analysis, with none being RCTs.

### 3.1. Characteristics of Reviewed Studies

Among the included studies, two were prospective studies [13,17], and the remaining twenty-three were retrospective [4,7,9,11,12,14,18,19,20,21,22,23,24,25,26,27,28,29,30,31,32,33,34,35]. Among the 25 studies examined, 11 [9,12,14,17,18,21,23,25,26,27,35] included comparative analyses involving different locoregional treatment groups, while 12 [4,13,17,18,19,20,27,28,30,31,33,35] did not provide data on SLND, and 1 did not report data on RT [34]. In cases where information on SLND or RT was not reported, data were selectively extracted from the available treatment arms.

The publications included in the network meta-analysis provided research spanning more than two decades from 2001. The study population comprised breast cancer patients with sISLM. The sample sizes of included studies varied, ranging from 33 to 1827, resulting in a pooled cohort of 3346 patients. Among these, 760 patients were treated with SLND + RT + ST, while 2061 patients underwent RT + ST, 13 were treated with SLND + ST, 11 received only RT, and 494 received only ST. Follow-up periods ranged from 24 to 105 months across the included studies, with most studies having median follow-up durations between 36 and 75 months (Table 1).

In terms of further treatment details, RT doses ranged from 45 to 66 Gy, typically targeting the supraclavicular fossa and axilla, with common regimens including 50 Gy in 25 fractions or a total dose of up to 60–66 Gy with a boost, as described in several included studies [9,14,17,18,21,25,26,35]. Where ST details were reported, most patients received anthracycline- and taxane-based chemotherapy regimens as part of their neoadjuvant or adjuvant treatment protocols [9,14,17,18,21,23,25,26]. HER2-positive patients were treated with trastuzumab in studies that reported targeted therapy use [9,17,21,26]. However, information on the duration of HER2-directed therapy or endocrine treatment was inconsistently reported or omitted entirely in several of the reviewed studies.

### 3.2. Primary Analysis

The reconstruction of individual patient datasets for use in the network meta-analysis was performed for the following outcomes: OS and DFS. Of the 10 included studies, 10 published Kaplan–Meier curves for OS with a pooled cohort of 3346 [9,14,17,18,21,23,25,26,27,35] (Figure 2). Notably, patients within the “RT + ST” treatment arm were chosen as the reference group to serve as a common comparator. The pooled hazard ratio (HR) for “SLND + RT + ST” versus “RT + ST” was calculated as 0.97 (95% CI: 0.63, 1.48). Similarly, the pooled HR for “SLND + ST” versus “RT + ST” was 0.91 (95% CI: 0.24, 3.41). Regarding ST alone versus RT + ST, the pooled HR for OS was 1.74 (95% CI: 0.88, 3.44), with all confidence intervals (CIs) crossing 1, suggesting that the differences in OS among the various treatment groups were statistically insignificant.

Six studies published Kaplan–Meier curves for DFS, with a pooled cohort of 1413 patients [9,12,14,23,25,26] (Figure 3). The pooled HR for “SLND + RT + ST” vs. “RT + ST” was calculated at 0.95 (95% CI: 0.66, 1.36), with CIs crossing 1, suggesting that there was no statistically significant difference in the DFS between with and without additional neck dissection.

### 3.3. Sensitivity Analyses

Sensitivity analyses assessing OS were conducted for SLND, categorized by the level of nodal dissection, and for RT, categorized by the median radiation dose. These subgroup analyses were undertaken to account for variations in the extent of neck dissection among patients who underwent SLND and discrepancies in the dosages of RT administered to patients.

Neck dissection levels were stratified into strictly SLND (level V only) and extended neck dissection (level V+), with the level of dissection available in eight studies [9,14,18,21,23,25,26,27]. When comparing studies reporting level V-only dissection [21,23], pooled HR for SLND + RT + ST was 0.47 (95% CI: 0.29, 0.77) vs. RT + ST with maximal homogeneity (I2 = 0%) (Figure 4).

When evaluating studies that conducted extended neck dissection beyond level V [9,14,18,25,26,27], pooled HR for SLND + RT + ST was 1.41 (95% CI: 1.10, 1.80) vs. RT + ST. ST alone had a statistically significant higher HR (2.13, 95% CI: 1.87, 2.42) vs. RT + ST (Figure 5).

Sensitivity analysis was also performed evaluating the impact of RT dose on OS, with eight studies reporting the median RT dose [9,14,17,18,21,25,26,35]. Studies were stratified according to lower RT doses (45–50 Gy) and higher RT doses (55–65 Gy). For doses ranging from 45 to 50 Gy [14,18], “SLND + RT + ST” exhibited a higher HR of 1.61 (95% CI: 1.09, 2.38) compared to “RT + ST” alone. Conversely, at higher RT doses of 55–65 Gy [9,17,21,25,26,35], pooled HR for “SLND + RT + ST” was 0.94 (95% CI: 0.54, 1.64) vs. “RT + ST” alone (Figure 6), with CIs crossing 1, suggesting that there was no statistically significant difference in the OS between with and without neck dissection.

### 3.4. Risk-of-Bias Assessment

We employed the Risk of Bias In Non-randomized Studies of Interventions (ROBINS-I) to assess the studies’ quality [36]. Quality control for included studies was performed independently by two reviewers to reduce bias. The figure displays the risk of bias analysis of the studies and presents their conclusions as a percentage across all included studies for each risk of bias domain. As detailed in Figure 7, most studies were rated as having low-to-moderate risk of bias across various domains, with no studies deemed to have a high risk of bias.

### 3.5. GRADE Assessment

The overall quality of evidence for OS was rated low due to the retrospective nature of the included studies, which introduced risk of bias, and the lack of direct randomized comparisons. Imprecision in effect estimates, with wide confidence intervals crossing 1, further contributed to the downgrading of evidence. The GRADE assessment for DFS was rated as moderate, as there was less heterogeneity in this outcome compared to OS, but the overall quality was still affected by the non-randomized design of most studies and variability in treatment protocols.

## 4. Discussion

This review synthesized findings from 25 studies investigating locoregional treatment strategies for breast cancer patients with sISLM. Among these, 10 studies contributed to the analysis of OS, encompassing 3346 patients, and 6 studies assessed DFS, with a pooled cohort of 1413 patients. The network meta-analysis found that SLND combined with RT and ST demonstrated comparable DFS and OS outcomes to RT and ST alone. Sensitivity analyses revealed that level V-only dissections provided significant survival benefits (HR: 0.47, 95% CI: 0.29–0.77), whereas more extensive dissections beyond level V were associated with worse outcomes (HR: 1.41, 95% CI: 1.10–1.80). These results suggest that, while SLND may offer benefits in specific scenarios, its broader application requires careful consideration, particularly given the associated morbidity and the role of RT dose intensity in influencing survival.

The role of SLND in breast cancer patients with sISLM remains a subject of debate. With improvement in diagnoses and multidisciplinary approaches and the reclassification of sISLM to stage N3C in AJCC 6th edition, sISLM is no longer viewed as a distant metastasis. This shift in classification creates the possibility of curative treatment through combined-modality therapy, including NAC, targeted therapy, surgery, RT, and endocrine therapy [37,38,39]. SLND combined with RT + ST appears to provide comparable outcomes to RT + ST alone, with level V-only dissection offering superior survival benefits compared to more radical neck dissections.

Patients with sISLM breast cancer usually receive radical or modified mastectomy, although some receive breast-conserving surgery [40]. The current NCCN guidelines recommend total or partial mastectomy with the axillary dissection of axillary levels I and II, including level III dissection if palpable nodes are present. In cases where disease is found in the infraclavicular and supraclavicular regions, the guidelines advocate for management through locoregional therapy [6]. Although studies report that locoregional therapy with radiation therapy can achieve locoregional control rates exceeding 80%, these therapies are associated with much lower 5-year OS and DFS, which range from 33.3% to 47% for OS and 25% to 34% for DFS [4,12,20,32,33,41]. This phenomenon may be attributed to residual tumor presence leading to tumor development and growth, often because the RT dose is unable to achieve the radical threshold, a limitation imposed by the presence of surrounding vital organs and neurovascular structures, such as the brachial plexus nerve. Unfortunately, in this review, detailed RT protocols (e.g., fields, GTV, elective volume definitions) were not uniformly available across all studies. Where reported, RT doses ranged from 45 to 66 Gy, typically targeting the supraclavicular fossa and axilla, with common regimens including 50 Gy in 25 fractions or a total dose of up to 60–66 Gy with a boost, as described in several included studies [9,14,17,18,21,25,26,35]. The current RT protocols are further constrained by the need to balance treatment efficacy with the risk of radiation-induced injury, such as radiation-induced brachial plexopathy (RIBP), which can lead to conditions like allodynia and movement disorders in the affected limb [42]. Moreover, complications such as radiation dermatitis and radiation pneumonitis, commonly associated with higher RT doses, can significantly diminish patients’ quality of life [43].

The surgical management of the SLN region offers a promising curative option, in which SLND may directly reduce tumor load, preventing the spread of tumor cells and providing a more precise assessment of lymph node status [9]. However, the unique anatomical position of the SLN and the lack of a standardized protocol for the extent of lymph node dissection have resulted in sporadic and unsystematic studies on aggressive locoregional surgery for sISLM. The extent of supraclavicular neck dissection varied between included studies. In the context of breast cancer, the term “level V dissection” corresponds to dissection of the posterior cervical triangle—posterior to the sternocleidomastoid muscle—consistent with conventional head and neck anatomical definitions [44]. However, the precise nodal levels dissected (e.g., inclusion of level III or IV) were inconsistently defined across studies. To address this, we stratified the sensitivity analysis based on whether the original study authors described the procedure as being limited to level V or as an extended neck dissection involving multiple levels. In head and neck cancer surgery, the supraclavicular region is relative to the cervical IV (inferior jugular vein) and V region (posterior cervical triangle). The higher position of III and II (upper and middle segments of jugular vein) is excluded in the drainage area of a conventional mammary gland based on previous mapping studies [25]. All lymph node metastases higher than the superior jugular vein fall to M1, and this provides a reference for the clinical dissection of supraclavicular lymph nodes. Although some studies include cervical III, IV, and V lymph nodes [26] in their dissection. Furthermore, the adequate number of nodes to be harvested is arbitrary. In the study of Song et al. [25] wherein mainly level IV and Vb neck dissection were performed, the average number of lymph node harvested was nine.

Another important limitation is the inconsistent reporting of axillary lymph node status across the included studies. Axillary involvement is a well-established prognostic factor in breast cancer, with higher axillary nodal burden associated with significantly poorer survival outcomes [45]. For instance, patients with ≥10 positive axillary nodes have been shown to have markedly reduced 5-year DFS compared to those with limited nodal involvement [26,46]. The lack of uniform data on axillary disease burden limited our ability to adjust for this variable in the analysis or stratify outcomes accordingly. This is particularly relevant given that more extensive supraclavicular dissection may have been performed in patients with more widespread regional disease, potentially confounding survival outcomes.

Consequently, there is a lack of clear recommendations in the NCCN guidelines regarding this approach. The main hypothesis revolves around whether more extensive surgery can enhance local control, prevent distant metastasis, and ultimately prolong survival outcomes for breast cancer patients with sISLM. In our NMA, we performed sensitivity analyses to categorize the level of SLND into two groups: strictly level V dissection, in which only the SLN is dissected, and extended neck dissection, in which further cervical lymph node levels were dissected. When unstratified, our systematic review and NMA, which obtained evidence from 11 non-randomized studies, suggested that combined therapy in the form of SLND and targeted RT to the supraclavicular region had similar OS and DFS to RT alone. However, further analysis comparing the level of neck dissection revealed that strictly level V dissection conferred a significant advantage in terms of OS when compared to RT alone (HR: 0.47; 95% CI: 0.29, 0.77), while extended neck dissection had poorer OS when compared to RT alone (HR: 1.41; 95% CI: 1.10, 1.80). The differences in results could be due to studies in which patients undergoing extended neck dissection could have had extended LN metastases, which predicts poorer outcome and survival [47].

The opponents of SLND often point to its associated morbidity, which includes an 80% probability of having a limited range of motion of head, neck, and shoulder function, 34% incidence of neuromuscular sensorial changes, 4% risk of lymphatic leakage, and 19% chance of lymphatic reflux [22,48]. The effectiveness of RT has also been reported, where Song et al. found that surgery offered no benefit over RT alone [25], and Diao et al. reported high 5-year DFS of 83% with RT alone in patients with residual SCV node size of more than one centimeter post-NAC [11]. Given the morbidity linked with radical neck dissections and the efficacy of RT, clinicians may advocate for RT as the primary locoregional treatment for sISLM. Therefore, the decision for SLND is often made on a case-by-case basis particularly when persistent disease is localized to the supraclavicular region, which impacts the ability to discontinue systemic therapy.

With the current understanding of tumor biology, many factors are likely to have affected the DFS and OS of patients with N3C disease beyond SLND alone. The molecular subtype of tumors and the response to neoadjuvant chemotherapy influence DFS and OS. The rate of supraclavicular pathologic complete response (pCR) itself was reported by Zhu et al. to be 53.7% [7]. Triple-negative disease and HER2-positive disease were found to be associated with supraclavicular pCR in the study. The predictive value for DFS was found to be more pronounced in patients with supraclavicular pCR than in those with breast pCR (HR: 0.48, 95% CI: 0.23–1, *p* = 0.05) or axillary pCR (HR: 0.96, 95% CI: 0.44–2.09, *p* = 0.92) [7]. These findings were supported by Diao et al., which reported that patients who achieved ypN0 compared to those who did achieve it not had a higher 5-year OS of 84% vs. 66% (*p* = 0.01) [11]. Patients who had a primary tumor pathologic complete response (ypT0) vs. those who did not have it had a 5-year OS of 76% vs. 71% (*p* = 0.33) [11]. The estrogen receptor (ER) status is also an independent factor determining DFS and OS. In the study of Sun et al., they found statistical significance (*p* = 0.003) in the 5-year DFS rate of ER-positive patients, 43.6%, compared to ER-negative patients, 18.5%, as well as for the 5-year OS (*p* < 0.01) of ER-positive patients, 89.4%, compared to ER-negative patients [26]. Similarly, Diao et al. observed significant differences in OS based on the ER status, with a 5-year OS of 78% for ER+/HER2− patients and 57% for ER−/HER2− patients (*p* = 0.002) [11]. These findings suggest that patients with a negative ER status and iSLM may have a poorer prognosis and could benefit from more aggressive treatment. In this regard, artificial intelligence (AI) models could be used to aid careful patient selection and multidisciplinary planning for optimal outcomes [49].

The presence of axillary lymph node metastases is also an independent factor influencing 5-year DFS in patients with ISLM breast cancer. Historically, it was theorized that the axillary metastasis route is through each level of the lymph node station. Thus, the more the positive lymph nodes for metastasis [2], the poorer is the patient’s prognosis. The 5-year DFS rate of patients with positive axillary lymph node ≥ 10 was found to be 5.3% as compared to 47.8% for patients with less than ten positive axillary lymph nodes, *p* = 0.005 [26], and the difference was statistically significant.

This NMA has several notable strengths, including its comprehensive synthesis of two decades of research and its novel use of sensitivity analyses to examine the impact of SLND extent and RT dose. However, several limitations must be acknowledged. First, all included studies were retrospective in nature, introducing inherent risks of selection bias and confounding. The notable absence of RCTs limits the strength and generalizability of our conclusions. Second, there was substantial heterogeneity across studies in terms of treatment protocols, including the extent of SLND, RT fields and dosages, systemic therapy regimens, and timing of interventions. Third, important prognostic variables—such as axillary lymph node burden and tumor molecular subtype (e.g., ER, PR, HER2 status)—were not consistently reported or controlled for, limiting our ability to adjust for these factors in the analysis. This is especially relevant given that both higher axillary nodal burden and hormone receptor negativity are independently associated with poorer survival outcomes, regardless of surgical extent [11,26]. Fourth, the definitions of SLND levels varied across studies, which may affect the reproducibility of our level V versus extended dissection subgroup findings. While we attempted to account for these variations through stratified sensitivity analyses, residual confounding remains possible. Finally, follow-up durations varied widely across studies, and the lack of granular data precluded analyses of recurrence patterns or quality-of-life outcomes. Despite these limitations, our findings offer valuable insights and hypotheses to inform future research. Ongoing randomized trials (ChiCTR1900023098 and NCT03716245) [50,51] are expected to provide more definitive guidance on the role of SLND in patients with sISLM.

## 5. Conclusions

RT combined with systemic therapy (RT + ST) remains the standard of care for patients with breast cancer presenting with sISLM. Our findings suggest that the addition of SLND, when limited to level V nodes, may offer a survival advantage in carefully selected patients (e.g., those with isolated level V supraclavicular involvement, who achieve partial response to systemic therapy, without extensive axillary or mediastinal involvement, and without signs of aggressive or rapidly progressing disease). In contrast, more extensive neck dissections appear to be associated with poorer outcomes, likely reflecting more aggressive disease biology rather than the surgical intervention itself. However, the impact of molecular subtypes on disease-free and overall survival could not be fully accounted for in this study. Higher RT doses (≥60 Gy) also appeared to favor improved outcomes. Given the retrospective design of all included studies, the findings should be interpreted with caution. While level V dissection appears promising, recommendations should remain tentative. Future RCTs are needed to confirm these findings and refine clinical practice guidelines for breast cancer patients with sISLM.

## Figures and Tables

**Figure 1 cancers-17-02081-f001:**
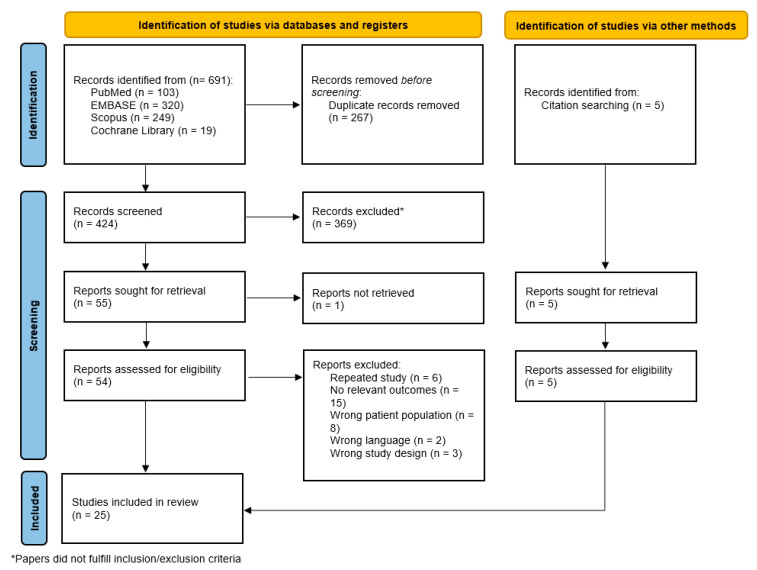
PRISMA flowchart showing the study selection process.

**Figure 2 cancers-17-02081-f002:**
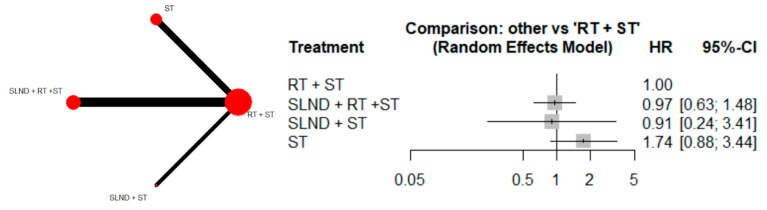
Forrest plot comparing OS across SLND + RT + ST vs. SLND + ST vs. ST alone with RT + ST as reference group.

**Figure 3 cancers-17-02081-f003:**
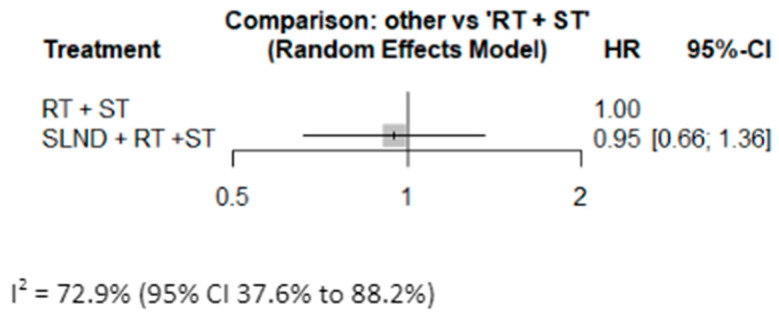
Forrest plot comparing DFS between SLND + RT + ST vs. RT + ST.

**Figure 4 cancers-17-02081-f004:**
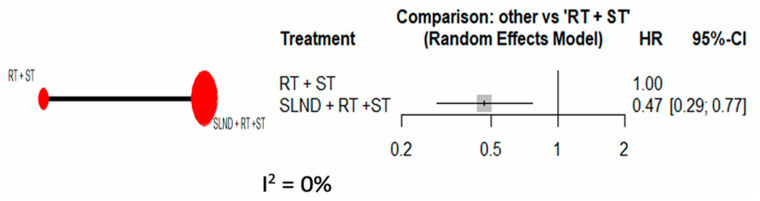
Forrest plot comparing OS between SLND (level V-only) + RT + ST vs. RT + ST.

**Figure 5 cancers-17-02081-f005:**
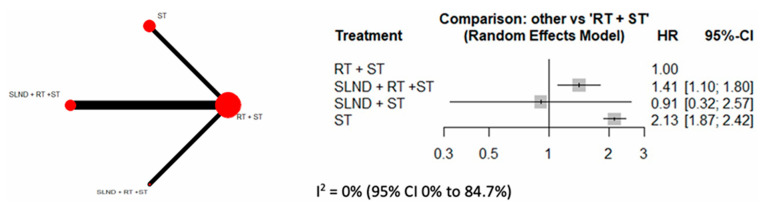
Forest plot comparing OS across extended neck dissection + RT + ST vs. SLND + ST vs. ST alone with RT + ST as reference group.

**Figure 6 cancers-17-02081-f006:**
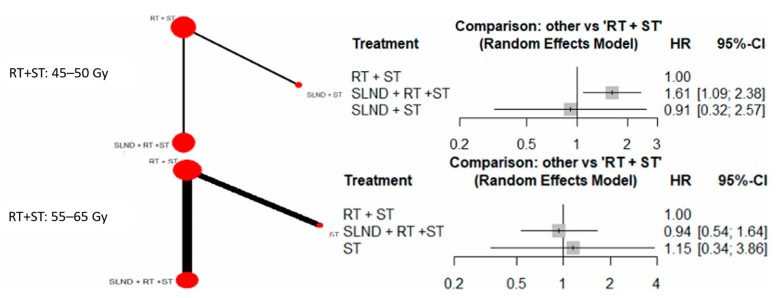
Sensitivity analysis stratified by RT dose, evaluating OS across treatment modalities with RT + ST as the reference.

**Figure 7 cancers-17-02081-f007:**
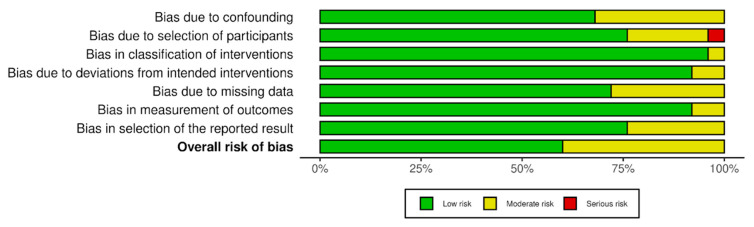
Summary of risk of bias of reviewed studies.

**Table 1 cancers-17-02081-t001:** Baseline characteristics of studies included in this review.

Study		Ai et al. (2020) [14]	Chang et al. (2013) [18]	Jung et al. (2015) [21]	Kim et al. (2020) [12]	Lv et al. (2021) [23]	Pergolizzi et al. (2001) [17]	Song et al. (2023) [25]	Sun et al. (2020) [26]	Tamirisa et al. (2021) [27]	Feng, et al. (2023) [9]	Fan et al. (2010) [35]
**Study Characteristics**	Country	China	China	Korea	Korea	China	Italy	China	China	USA	China	China
	Study Design	RS	RS	RS	RS	RS	Non-Randomized CT	RS	RS	RS	RS	RS
	Median Age, Years	48	47	49	49	50	ST only: 58 RT + ST only: 55	51	NR	60	57	45
	Age Range	25–83	28–69	27–81	27–80	22–77	ST only: 32–71 RT + ST only: 39–77	26–70	NR	50–69	24–75	24–74
	Median Follow-up, Months	36	47	NR	72	24	105	SLND + RT + ST: 53.7 RT + ST: 63.5	75	NR	71	93
**Locoregional Treatment Group**	Total Population	305	29	111	104	353	37	293	108	1827	250	33
	SLND + ST + RT	146	0	73	57	307	0	85	84	0	65	0
	RT + ST	159	16	16	47	46	19	208	24	1362	185	26
	SLND + ST	0	13	0	0	0	0	0	0	0	0	0
	RT Only	0	0	11	0	0	0	0	0	0	0	0
	SLND only	0	0	0	0	0	0	0	0	0	0	0
	ST Only	0	0	4	0	0	18	0	0	465	0	7
	No Treatment	0	0	Unknown: 7	0	0	0	0	0	0	0	0
**SLND**	Levels	III, IV, Vb	IV, V	Vb	NR	Vb	NA	IV, Vb	III, IV, V	NA	I, II, III, IV, V	NA
**Radiotherapy**	Median Dosage (Gy)	NR	NR	60	50	NR	60	60	60	NR	60	60
	Range	46–50	45–50	45–66	45–64.8	NR	NR	50–70	NR	NR	NR	NR
**Molecular Subtype**	ER+, n	198	19	53	105	202	11	188	60	NR	141	NR
	PR+ (%)	166	19	53	66	176	NR	188	60	NR	NR	NR
	HER2+ (%)	83	10	48	NR	156	NR	113	48	NR	93	NR
	Ki67 < 30	NR	NR	NR	NR	51	NR	87	22	nr	NR	NR
	Ki67 > 30	NR	NR	NR	0	302	NR	188	86	NR	NR	NR
**Primary Tumor Size**	T0 (%)	0	0	NR	56	NR	0	NR	0	NR	0	0
	T1 (%)	57	2	NR	78	NR	3	NR	15	NR	47	9
	T2 (%)	195	11	NR	20	NR	18	NR	71	NR	116	11
	T3 (%)	29	10	NR	4	NR	6	NR	NR	NR	51	8
	T4 (%)	24	6	NR		NR	10	NR	NR	NR	36	5

Abbreviations: RS: Retrospective Study; CT: chemotherapy; NR: not reported; ST: Surgery Treatment; RT: radiotherapy; SLND: Sentinel Lymph Node Dissection; SLND + ST + RT: Sentinel Lymph Node Dissection + Surgery Treatment + radiotherapy; RT + ST: radiotherapy + Surgery Treatment; SLND + ST: Sentinel Lymph Node Dissection + Surgery Treatment; ER+: Estrogen Receptor Positive; PR+: Progesterone Receptor Positive; HER2+: Human Epidermal Growth Factor Receptor 2 Positive.

## Data Availability

This is a systematic review of published studies; no new data were created or analyzed in this study.

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
