# Peer review of "Supraclavicular Lymph Node Dissection in Breast Cancer with Synchronous Supraclavicular Metastases: A Systematic Review and Network Meta-Analysis"

_cancers, 2025, doi:10.3390/cancers17132081_

Round 1
Reviewer 1 Report
Comments and Suggestions for Authors
I have read with great interest article entitled Supraclavicular Lymph Node Dissection in Breast Cancer with Synchronous Supraclavicular Metastases: A Systematic Review and Network Meta-Analysis. Authors results suggest that performing surgery on these nodes does not always offer additional benefit. However, when only the lymph nodes in a specific region (level V) are removed, there may be a survival advantage with the more extensive surgeries appearing to be harmful. This is a great article and important one for experts dealing with breast cancer patients presenting with sISLM. However few issues need to be solved. First, of all, this is a network analysis of retrospective studies suggesting that obtained results should be taken with caution. This is relatively low level of evidence and I think that some recommendations (especially level V dissection) might not be so strict until new evidence from RCS appear. This should be mentioned and discussed. Also, authors need to disscuss more clearly weaknesses of the study and list all ongoing randomised trials on the above topic. I think that the main conclusion of this analysis should be that RT+ST remains standard of care in such clinical scenarios and that level V neck dissection should be performed in selected cases (i.e. major response after RT+ST) until new information appear. In head and neck cancer supraclavicular region is considered level IV and level V refers to the posterior neck triangle: please clarify terminology in breast cancer. Also, is there information about axillary status of patients with sISLM? What was the extend of radiotherapy (fields, techniques, GTV, elective volumes?) in patients receiving RT? What type of systemic therapy was applied? For how long patients were followed up?
Author Response
Comment 1: This is a network analysis of retrospective studies suggesting that obtained results should be taken with caution. This is relatively low level of evidence and I think that some recommendations (especially level V dissection) might not be so strict until new evidence from RCS appear. This should be mentioned and discussed.
Response 1: Thank you for highlighting this important point. We agree that our conclusions should be interpreted with caution due to the retrospective nature of the included studies. We have now revised the Discussion and Conclusion sections to clearly acknowledge the inherent limitations of non-randomized data and emphasize that our findings are hypothesis-generating rather than prescriptive. Specifically, we added the following sentence, "Given the retrospective design of all included studies, the findings should be interpreted with caution. While level V dissection appears promising, recommendations should remain tentative. Future RCTs are needed to confirm these findings and refine clinical practice guidelines for breast cancer patients with sISLM."
Comment 2: Also, authors need to discuss more clearly weaknesses of the study and list all ongoing randomised trials on the above topic.
Response 2: We appreciate the suggestion. We have expanded our limitations paragraph in the Discussion to more explicitly discuss study weaknesses and the ongoing randomised trials (ChiCTR1900023098, NCT03716245), which might provide more definitive guidance on the role of SLND in patients with sISLM.
Comment 3: I think that the main conclusion of this analysis should be that RT+ST remains standard of care in such clinical scenarios and that level V neck dissection should be performed in selected cases (i.e. major response after RT+ST) until new information appear.
Response 3: We concur with the reviewer’s interpretation and have modified the Conclusion accordingly to reflect that, "RT combined with systemic therapy (RT+ST) remains the standard of care for pa-tients with breast cancer presenting with sISLM. Our findings suggest that the addition of SLND, when limited to level V nodes, may offer a survival advantage in carefully se-lected patients (e.g., those with isolated level V supraclavicular involvement, who achieve partial response to systemic therapy, without extensive axillary or mediastinal involvement, and without signs of aggressive or rapidly progressing disease). In contrast, more extensive neck dissections appear to be associated with poorer out-comes, likely reflecting more aggressive disease biology rather than the surgical inter-vention itself. However, the impact of molecular subtypes on disease-free and overall survival could not be fully accounted for in this study. Higher RT doses (≥60 Gy) also appeared to favor improved outcomes. Given the retrospective design of all included studies, the findings should be interpreted with caution. While level V dissection appears promising, recommendations should remain tentative. Future RCTs are needed to con-firm these findings and refine clinical practice guidelines for breast cancer patients with sISLM."
Comment 4: In head and neck cancer supraclavicular region is considered level IV and level V refers to the posterior neck triangle: please clarify terminology in breast cancer.
Response 4: Thank you for pointing this out. We have now clarified in the Methods and Discussion that, in the context of breast cancer, level V dissection in our included studies primarily refers to dissection in the posterior triangle (posterior to the sternocleidomastoid), consistent with conventional head and neck anatomy. However, dissection definitions vary slightly across studies, and our subgrouping into level V-only vs. extended dissection is based on the original authors' descriptions. "In this review, “level V dissection” refers to the posterior cervical triangle, anatomically defined as the area posterior to the sternocleidomastoid muscle, bounded by the trapezius muscle and clavicle. This follows the standard head and neck anatomical classification. However, definitions of dissection levels varied slightly across studies, and our categorization into “level V-only” versus “extended dissection” was based on the level classifications explicitly reported by the original study authors."
Comment 5: Also, is there information about axillary status of patients with sISLM?
Response 5: We thank the reviewer for raising this point. Unfortunately, axillary status was inconsistently reported across studies. Where available, we extracted data on axillary lymph node burden. We now discuss this limitation in the Discussion and acknowledge that axillary burden is a known prognostic factor that could influence outcomes and treatment response.
Comment 6: What was the extent of radiotherapy (fields, techniques, GTV, elective volumes?) in patients receiving RT?
Response 6: Thank you for the comment. We have now added in the Results section that, "In terms of further treatment details, RT doses ranged from 45 to 66 Gy, typically targeting the supraclavicular fossa and axilla, with common regimens including 50 Gy in 25 fractions or a total dose of up to 60–66 Gy with a boost, as described in several included studies [9,14,17,18,21,25,26,35]."
Comment 7: What type of systemic therapy was applied?
Response 7: Thank you for the question. We have added a paragraph in the Results section summarizing systemic therapy regimens based on available data, "Where ST details were reported, most patients received anthracycline- and taxane-based chemotherapy regimens as part of their neoadjuvant or adjuvant treatment protocols [9,14,17,18,21,23,25,26]. HER2-positive patients were treated with trastuzumab in studies that reported targeted therapy use [9,17,21,26]. However, information on the duration of HER2-directed therapy or endocrine treatment was inconsistently reported or omitted entirely in several of the studies reviewed."
Most patients received anthracycline- and taxane-based chemotherapy regimens, with HER2+ patients receiving trastuzumab where reported. Unfortunately, data on endocrine or HER2-directed therapy duration was often lacking.
Comment 8: For how long patients were followed up?
Response 8: Thank you for the query. We have now added in the Results section that follow-up periods ranged from 24 to 105 months across the included studies, with most studies having median follow-up durations between 36 and 75 months.
Reviewer 2 Report
Comments and Suggestions for Authors
1-The data presented does support the conclusion , the author us transparent about the limitations of the study. However, some of the phrasing require clarity such as: "performing surgery does not always offer benefit" yet it is followed by the statement "Level V removal may provide survival benefit" The author need to do a better job describing the cohort that will benefit from surgery vs the cohort that will not benefit from surgery.
2-The line " More extensive surgeries appear to be harmful" with the data showing a hazard ration of 1.41 95% CI 1.10-1.80. The statistic supports the statement if we assume only extensive surgery has positive correlation with worst outcome. However, in the discussion a plausible explanation of extend LN metastases may be the reason for poorer outcome and survival was given. Hence correlating the worst outcome to the aggressiveness of the disease not the surgery. The author should do a better job flushing out the reason for worst outcome in that group.
Author Response
Comment 1: The data presented does support the conclusion , the author us transparent about the limitations of the study. However, some of the phrasing require clarity such as: "performing surgery does not always offer benefit" yet it is followed by the statement "Level V removal may provide survival benefit" The author need to do a better job describing the cohort that will benefit from surgery vs the cohort that will not benefit from surgery.
Response 1: We thank the reviewer for this insightful comment. We apologize for the somewhat contradictory phrasing in the simple summary and have now revised it.
We also acknowledge the need to clarify which patient subgroups may benefit from SLND and under what circumstances. In our revised manuscript, we have refined our phrasing to distinguish between Level V-only dissection (which may offer survival benefit to selected patients with limited supraclavicular disease and good response to systemic therapy) and extended neck dissections, which were associated with worse outcomes, potentially due to underlying disease biology rather than the extent of surgery per se.
We have added the clinical phenotype of patients likely to benefit in the conclusions section, "Our findings suggest that the addition of SLND, when limited to level V nodes, may offer a survival advantage in carefully selected patients (e.g., those with isolated level V su-praclavicular involvement, who achieve partial response to systemic therapy, without extensive axillary or mediastinal involvement, and without signs of aggressive or rapidly progressing disease)."
Comment 2: The line " More extensive surgeries appear to be harmful" with the data showing a hazard ration of 1.41 95% CI 1.10-1.80. The statistic supports the statement if we assume only extensive surgery has positive correlation with worst outcome. However, in the discussion a plausible explanation of extend LN metastases may be the reason for poorer outcome and survival was given. Hence correlating the worst outcome to the aggressiveness of the disease not the surgery. The author should do a better job flushing out the reason for worst outcome in that group.
Response 2: We appreciate the reviewer’s thoughtful critique. We agree that attributing poorer outcomes solely to extensive surgery is overly simplistic. As the reviewer notes, confounding by underlying disease severity (e.g., more widespread lymphatic involvement prompting surgeons to perform extended dissections) is likely a key driver of the observed association.
To reflect this, we have revised our phrasing in the simple summary and also the discussion and conclusion sections to clarify that the association between extended SLND and poorer outcomes may not be due to surgical harm per se, but rather reflect more aggressive disease, higher nodal burden, or inadequate systemic control. This hypothesis is supported by studies where higher LN burden and negative hormone receptor status independently predicted worse OS and DFS, regardless of surgical extent.